# Differential Associations Between Adaptability and Mental Health Symptoms Across Interpersonal Style Groups: A Network Comparison Study

**DOI:** 10.3390/bs15101307

**Published:** 2025-09-25

**Authors:** Shixiu Ren

**Affiliations:** Tianjin Academy of Educational Sciences, No. 25 Fukang Road, Nankai District, Tianjin 300191, China; shixiuren@mail.bnu.edu.cn

**Keywords:** interpersonal style, adaptability, mental health, network comparison, college students

## Abstract

The university period is a transitional stage during which students develop heterogeneous interpersonal styles to navigate complex social demands. While prior studies have linked interpersonal functioning to adaptability and mental health, structural differences across interpersonal style groups remain underexplored. Therefore, the current research was designed to examine whether and how adaptability is differentially related to mental health symptoms when considered within the framework of distinct interpersonal style profiles. Using K-means clustering, we identified three distinct interpersonal profiles: the withdrawn and avoidant type, the overinvolved and compliant type, and the well-adjusted interpersonal type. Based on this classification, network analyses were conducted to examine how six dimensions of adaptability related to three core mental health symptoms within each group. The results showed a consistent pattern across all profiles, with emotional adaptability negatively associated with depression, anxiety, and stress. Subsequent network comparison analyses demonstrated that the withdrawn and avoidant group differed significantly in structure from the well-adjusted interpersonal group, particularly in the connections involving emotional, interpersonal, and economic adaptability. By uncovering meaningful differences in adaptability-mental health associations across interpersonal style, this study provides a foundation for designing targeted strategies that address the unique adaptabilities and mental health problems of distinct interpersonal profiles.

## 1. Introduction

The university period constitutes a distinct and pivotal developmental phase characterized by substantial personal maturation and heightened social complexity ([7]). During this period, students encounter increasingly intricate interpersonal demands, navigating diverse relationships including peer friendships, romantic partnerships, academic collaborations, and interactions with faculty and administrative personnel ([34]). In adapting to and managing these varied relational contexts, individuals typically develop consistent and characteristic patterns of interpersonal behavior, conceptualized as interpersonal styles. [41] ([41]) proposed that interpersonal style refers to an individual’s recurrent patterns of interpersonal behavior within social contexts. Researchers have further identified several key dimensions shaping these patterns, including intrusive, vindictive, socially avoidant, nonassertive, cold, exploitable, overly nurturant, and domineering behaviors ([3]; [39]). When faced with complex and diverse interpersonal environments, college students tend to exhibit distinct interpersonal styles influenced by their personal experiences and psychological traits ([10]; [45]). For instance, [45] ([45]) conducted a latent profile analysis of interpersonal problem dimension scores from 469 undergraduate students in the Midwest and identified three distinct interpersonal profiles: flexible–adaptive, exploitable–subservient, and hostile–avoidant.

A growing body of research has underscored the strong association between college students’ interpersonal functioning and mental health ([36]; [27]). Mental health is commonly conceptualized as a multidimensional construct and a dynamic state of internal equilibrium that enables individuals to realize their abilities in harmony with the universal values of society ([19]). Within the university context, depression, anxiety, and stress are among the most prevalent and impairing forms of mental health, as they are closely linked to academic demands, social transitions, and identity development ([5]; [29]). Importantly, prior studies have shown that interpersonal functioning plays an important role in shaping these outcomes. For instance, [47] ([47]) found that students exhibiting high levels of social avoidance tend to report lower self-esteem and reduced interpersonal trust, which in turn contribute to elevated symptoms of depression and anxiety. In contrast, students who demonstrate adaptive interpersonal behaviors are more likely to experience fewer mental health problems ([11]; [38]).

Beyond its impact on mental health, interpersonal functioning also plays a vital role in shaping students’ adaptability to both college life and the broader social environment ([26]; [24]; [49]). Adaptability, in turn, is recognized as a crucial psychological capacity that enables students to adjust their thoughts, emotions, and behaviors in response to dynamic and uncertain circumstances ([13]; [28]). It encompasses multiple domains, including learning adaptability, professional adaptability, homesickness adaptability, interpersonal adaptability, emotional adaptability, and economic adaptability. When adaptability is low, maladaptive interpersonal patterns may further exacerbate adjustment difficulties. For example, [20] ([20]) reported that individuals characterized by excessive hostility or dependency often struggle with emotional maladjustment. Similarly, those with social avoidance tendencies are more likely to experience challenges such as homesickness and difficulties in interpersonal adjustment following college enrollment ([35]; [33]).

Although existing studies have differentiated distinct groups based on interpersonal styles and confirmed their heterogeneity ([45]; [40]), as well as demonstrated close interconnections among interpersonal styles, adaptability, and mental health ([36]; [49]), most of this research continues to rely primarily on traditional methods, such as regression and correlation analyses. While informative, these conventional approaches overlook potential variability in relationships between adaptability and mental health across individuals with distinct interpersonal patterns. Given the marked differences in interpersonal behaviors among individuals, associations between adaptability and mental health symptoms may also vary depending on interpersonal style. Consequently, there is still a lack of studies that compare how multiple dimensions of adaptability relate to diverse mental health symptoms across different interpersonal style groups. The Network Comparison Test (NCT) provides a powerful perspective for comparisons of variable structures across groups, thus enhancing the precision of psychological research on subgroup differences ([43]). Utilizing this methodology can more effectively clarify how adaptability and mental health symptoms uniquely relate to different interpersonal styles. Such insights are crucial for practitioners and educators in designing tailored interventions and preventive strategies that precisely address the specific psychological vulnerabilities and adaptive challenges linked to distinct interpersonal profiles, thereby enhancing students’ adaptability and mental health outcomes.

This study aims to clarify whether and how the associations between adaptability and mental health symptoms vary across interpersonal style groups by using network analysis. The analytical procedure comprises three steps. First, distinct subgroups with characteristic interpersonal styles are identified through cluster analysis. Next, subgroup-specific network models are estimated to depict the links between adaptability and mental health symptoms. The NCT is then used to test for structural differences across interpersonal style groups. This study contributes to the literature by integrating person-centered and network-based approaches to reveal how the associations between adaptability and mental health symptoms vary across distinct interpersonal style groups. The findings offer empirical support for the development of targeted mental health interventions, highlighting the critical role of interpersonal styles in shaping students’ adaptive functioning and psychological well-being.

## 2. Method

### 2.1. Study Design and Participants

We recruited a convenience sample of 405 undergraduate and graduate students from Tian, one of China’s largest cities. Participants were invited through classroom announcements and voluntarily took part in the study. The survey was administered in paper–pencil format and took approximately 5–8 min to complete. Data were collected in groups of about 90 individuals. To examine the extent of missing data, univariate statistical analyses were performed on each item. Results indicated that missing data rates ranged from 0.0% to 1.2%. As missingness was below the 10.0% threshold, and different handling methods did not yield significant differences ([30]), the expectation–maximization (EM) imputation method was applied. In total, 405 participants made up the final sample, with 70 males (20.8%) and 335 females (79.2%), with an average age of 19.520 years (*SD* = 1.932).

### 2.2. Setting

Participants were informed of the study purpose, confidentiality, and their right to withdraw at any point prior to data collection. Written informed consent was obtained from all participants before completing the survey. The study was reviewed and approved by the Human Research Protection Committee of Tianjin Normal University (ethical approval number: XL2020-08). Special attention was given to the administration of the Depression Anxiety Stress Scales (DASS-21), as it contains potentially distressing items. To minimize risks, participants were informed in advance that some questions might involve sensitive emotional content, but were reassured that there were no right or wrong answers. After completing the survey, all participants were debriefed to ensure they felt comfortable and safe.

### 2.3. Measurements

#### 2.3.1. The Inventory of Interpersonal Problems Circumplex Scale

The short version of the inventory of interpersonal problems circumplex scale, revised by [3] ([3]), is adapted from the original 127-item version ([23]). This shortened scale comprises 32 items distributed equally across eight dimensions: domineering, vindictive, cold, socially avoidant, nonassertive, exploitable, overly nurturant, and intrusive. An example item is “I always disclose my personal matters to others.” Respondents rate items on a 5-point Likert scale ranging from 1 (“extremely inconsistent”) to 5 (“extremely consistent”), with higher scores indicating more severe interpersonal problems. In this study, Cronbach’s α for the whole scale was 0.883, and the ω of the whole scale was 0.883. For the six dimensions, except for the exploitable dimension (Cronbach’s α = 0.554), the reliabilities of the other seven dimensions ranged from 0.609 to 0.787, which are generally considered acceptable ([4]). Despite the reliability of the exploitable dimension was below 0.600, prior research has shown that lower α values are common when the number of items is small, and such subscales can still be used meaningfully in research ([1]; [14]; [32]). The results of the CFA fit indices were good (CFI = 0.916, TLI = 0.901, SRMR = 0.052, RMSEA = 0.040, 90 percent C.I. of the RMSEA ranged from 0.035 to 0.046). That is, the reliability and validity of the scale were good.

#### 2.3.2. The Chinese Brief Version of the Depression, Anxiety, and Stress Scale

The Chinese adaptation of the depression-anxiety-stress scale (DASS-21) was developed by [21] ([21]). It consists of 21 items divided among three subscales: depression, anxiety, and stress (7 items each). Items, such as “I feel that my life has no meaning”, and rated on a 4-point Likert scale (1 = “Did not apply to me at all”, 4 = “Applied to me very much”. Reliability was strong (total α was 0.928; subscale α was [0.812, 0.858]; and total ω = 0.931). CFA confirmed a good fit (*CFI* = 0.922, *TLI* = 0.910, *SRMR* = 0.049, *RMSEA* = 0.067, 90 percent C.I. of the RMSEA ranged from 0.060 to 0.074).

#### 2.3.3. The Freshmen Adaptability Scale

The Freshmen Adaptability Scale in Chinese was adapted by [28] ([28]) from the original version by [13] ([13]). It assesses six domains of adaptability: learning, professional, homesickness, interpersonal, emotional, and economic, with four items in each dimension. An example item is “I found myself unprepared for the financial burden of university life.” Participants respond on a 6-point Likert scale, ranging from 1 (“strongly disagree”) to 6 (“strongly agree”). In the current sample, overall Cronbach’s α was 0.857, with subscale reliabilities between 0.752 and 0.888, and the total ω was 0.841. CFA showed satisfactory fit (CFI = 0.920, TLI = 0.905, SRMR = 0.060, RMSEA = 0.057, 90 percent C.I. of the RMSEA ranged from 0.051 to 0.063).

### 2.4. Statistical Methods

Initially, K-means clustering was applied via the package *cluster* in version of R 4.4.3 to identify distinct subgroups representing different interpersonal styles among college students. For each subgroup, a separate network depicting mental health and adaptability was estimated using Gaussian Graphical Models (GGMs) via the *qgraph* package in version of R 4.4.3 ([18]). In psychological network analysis, nodes represent variables and edges represent relationships between them ([15]). Sparse partial correlation matrices were derived using the Extended Bayesian Information Criterion Graphical Least Absolute Shrinkage and Selection Operator (EBICGLASSO) algorithm ([17]), with the tuning value λ fixed at 0.5 as recommended by prior research ([48]). Two centrality indices, namely strength and betweenness centrality, were used to assess the importance of each node within these networks ([42]). Strength centrality indicates how strongly a node is connected to others, reflecting its overall impact in the network. Betweenness centrality reflects how frequently a node serves as a bridge along the shortest paths connecting other nodes, thereby indicating its role in maintaining network cohesion. Finally, the NCT was employed to detect differences in the network structures of mental health and adaptability across the identified subgroups. The NCT evaluated invariance in global strength, overall network structure, and specific edges between groups ([43]), employing 2000 permutations. Global strength invariance examines differences in the cumulative connectivity of networks, with significant differences indicated by a *p*-value below 0.050. Network structure invariance evaluates the overall pattern of node interconnections, also using a significance threshold of *p* < 0.050. Single-edge invariance assesses differences in specific node relationships across groups, with a significant difference similarly indicated by a *p*-value below 0.050.

## 3. Results

### 3.1. Classification of Interpersonal Styles

Table 1 presents the within-cluster sum of squares (WCSS), Calinski–Harabasz (CH), Dunn, and Bayesian Information Criterion (BIC) indicators for different K-means clustering solutions, as determined using the Elbow method ([9]; [46]; [6]). According to the WCSS, it decreased with increasing numbers of clusters. The CH index favored a two-cluster solution, the Dunn index indicated that both two and three clusters were plausible, while the BIC supported three clusters. And we also conducted Latent Profile Analysis (LPA) as an alternative classification method, which likewise favored a three-class model, as shown below in Table A1 of the Appendix A. Taken together, these indices supported the three-cluster solution as providing the best balance between cohesion and separation while capturing meaningful heterogeneity in the data.

Figure 1 illustrates the mean scores across the dimensions of interpersonal problems for the three groups derived from the K-means clustering procedure. Cluster 1 (*n* = 135), labeled the “Withdrawn and Avoidant Type”, exhibited the highest scores on socially avoidant (*M* = 12.66), nonassertive (*M* = 12.30), and cold (*M* = 14.08), indicating a socially withdrawn and emotionally distant interpersonal style. Cluster 2 (*n* = 121), labeled the “Overinvolved and Compliant Type”, showed elevated scores on intrusive (*M* = 12.74), exploitable (*M* = 12.96), overly nurturant (*M* = 13.31), and nonassertive (*M* = 12.60), suggesting a pattern characterized by excessive self-disclosure, compliance, and a tendency to overextend in relationships. Cluster 3 (*n* = 149), labeled the “Well-Adjusted Interpersonal Type”, demonstrated comparatively lower scores across all dimensions, reflecting a more balanced and adaptive interpersonal style.

Beyond statistical fit, the three-cluster solution was also theoretically meaningful. The profiles closely resembled interpersonal style typologies frequently reported in previous research—namely, flexible–adaptive, exploitable–subservient, and hostile–avoidant ([45]). Moreover, they can be mapped onto established attachment theory ([8]): the well-adjusted interpersonal group corresponds to secure attachment, the overinvolved and compliant group reflects features of anxious–preoccupied attachment, and the withdrawn and avoidant group aligns with avoidant attachment. Thus, the retention of three clusters was justified by both empirical robustness and theoretical grounding, enhancing the interpretability and validity of the findings.

Furthermore, both the withdrawn and avoidant group and the overinvolved and compliant group had relatively high scores on the nonassertive dimension compared to the well-adjusted group. This finding suggested that low assertiveness might have been a common underlying feature of both overinvolved and withdrawn interpersonal patterns, highlighting it as a potential core difficulty shared by otherwise distinct maladaptive styles. Such a pattern underscored the importance of assertiveness in maintaining healthy interpersonal functioning and indicated that interventions targeting assertiveness skills may benefit individuals across different interpersonal profiles. Details can be seen in Figure 1.

### 3.2. Network Structure Comparison for Adaptability and Mental Health Symptoms

Three distinct network analyses were performed to investigate the relationship between six adaptability dimensions and three mental health symptoms across the identified group. Network stability was examined using the bootstrap approach available in the *bootnet* package. Specifically, a total of 1000 bootstrap iterations generated confidence intervals for the edge weights, providing an assessment of connection reliability. The results indicated that the connections between adaptability and mental health symptoms were stable and robust. For a visual representation of these findings, refer to Figure 2: the left panel displays the network for the withdrawn and avoidant group, the middle panel corresponds to the overinvolved and compliant group, and the right panel presents the network for the well-adjusted interpersonal type group.

Regarding network centrality, the withdrawn and avoidant group showed correlation stability coefficients of 0.593 (strength) and 0.052 (betweenness), indicating that node strength and betweenness remained correlated with the original data (*r* = 0.700) even after removing 59.3% and 5.2% of the sample. For the overinvolved and compliant group, the correlation stability coefficients were 0.438 (strength) and 0.207 (betweenness), while the well-adjusted interpersonal type group had 0.362 and 0.282. Following recommendations by [12] ([12]), strength centrality is recommended as a key indicator of node importance, and correlation stability coefficients above 0.250 suggests acceptable stability of [17] ([17]). Bootstrapped stability analyses confirmed that the networks were reliable for both college and middle school samples (see Figure 3).

As illustrated in Figure 4, the network linking six adaptability dimensions with three mental health symptoms. As shown in the figure, the associations between adaptability and mental health symptoms appeared largely similar across the three interpersonal style groups. For instance, emotional adaptability was negatively associated with depression, anxiety, and stress in all three groups. Specifically, the correlation coefficients ranged from −0.166 to −0.109 for students in the withdrawn and avoidant group, from −0.158 to −0.092 for the overinvolved and compliant group, and from −0.208 to −0.062 for the well-adjusted interpersonal type group. In addition, learning adaptability also showed a consistent negative correlation with depression across all groups. Further details can be found in Figure 4.

Moreover, Figure 5 illustrates the centrality indices of network nodes across the three interpersonal style groups. Across all groups, stress and emotional adaptability consistently exhibited high strength centrality, indicating their strong and direct associations with other variables in the networks. In terms of betweenness centrality, both depression and emotional adaptability demonstrated high values, suggesting their crucial bridging roles in the transmission of information within the psychological network structures of all groups. These findings highlight the central and connecting roles of emotional adaptability across multiple interpersonal styles, while also underscoring the persistent influence of stress and depression in the network of mental health and adaptability.

Furthermore, differences in the relationship between adaptability and mental health symptoms were observed across the three interpersonal style groups. To rigorously compare the network structures among the three groups, the NCT was employed, assessing the networks from three key dimensions: global strength, overall structure, and individual edge differences. It is important to note that the NCT can only compare two groups at a time; therefore, pairwise comparisons were conducted across the three interpersonal style groups. For the comparison between the withdrawn and avoidant group and the overinvolved and compliant group, the NCT results indicated no significant difference in network structure (*M* = 0.266, *p* = 0.135). Similarly, no significant difference was found between the overinvolved and compliant group and the well-adjusted interpersonal group (*M* = 0.186, *p* = 0.550).

However, a significant difference emerged when comparing the withdrawn and avoidant group with the well-adjusted interpersonal group (*M* = 0.320, *p* < 0.010), suggesting that at least one edge differed significantly between their network structures. Consequently, exploratory post hoc analyses were conducted on all edges to identify the specific connections responsible for the observed differences. The results revealed two significantly different edges between these groups. Specifically, the relationship between students’ interpersonal adaptability and emotional adaptability was significantly different (*E* = 0.221, *p* < 0.010), with no correlation observed in the withdrawn and avoidant group (*r* = 0.000), whereas a weak but significant negative correlation was found in the well-adjusted interpersonal group (*r* = −0.029, *p* < 0.050). In addition, a significant difference was found in the relationship between emotional adaptability and economic adaptability (*E* = 0.320, *p* < 0.001), with a small positive correlation in the withdrawn and avoidant group (*r* = 0.031, *p* < 0.050) and a strong positive correlation in the well-adjusted interpersonal group (*r* = 0.350, *p* < 0.001). Notably, no significant differences were observed between the two groups in the overall relationship patterns between adaptability and mental health symptoms.

## 4. Discussion

This study investigated how the interplay between adaptability and mental health symptoms varies across distinct interpersonal style groups among college students, leveraging the complementary strengths of person-centered classification and psychological network analysis. Using K-means clustering, we identified three distinct interpersonal profiles—withdrawn and avoidant, overinvolved and compliant, and well-adjusted interpersonal—each marked by unique patterns of interpersonal problems. These classifications provided a valuable foundation for exploring group-specific psychological dynamics. Despite notable differences in interpersonal styles, our findings revealed a remarkable degree of network invariance in the overall structure linking adaptability and mental health symptoms across the three groups. While subtle group differences emerged, these were confined to intra-adaptability connections—namely, the associations among different types of adaptability (e.g., emotional, interpersonal, and economic)—rather than the connections between adaptability and mental health symptoms. In other words, the associations between adaptability and psychological distress remained statistically invariant across groups, suggesting a stable and universal protective role of adaptability regardless of interpersonal style.

### 4.1. Interpersonal Style Profiles Among College Students

This study employed K-means clustering to classify college students into three distinct interpersonal style groups based on their scores on an interpersonal problem inventory scale: the withdrawn and avoidant type, the overinvolved and compliant type, and the well-adjusted type. Consistent with previous findings, students in the withdrawn and avoidant group exhibited high levels of coldness, social avoidance, and non-assertiveness, along with elevated scores on vindictiveness and exploitative tendencies ([2]). This profile reflects a pattern of interpersonal detachment, emotional distance, and diminished self-advocacy, which may hinder the formation of close relationships and prompt active distancing from peers due to fear of rejection or low perceived social competence. In contrast, the overinvolved and compliant group was characterized by excessive emotional investment in others, blurred personal boundaries, and a strong tendency to prioritize others’ needs over their own. Like prior research, this pattern reflects interpersonal overextension that, while superficially suggestive of social engagement, often results in emotional exhaustion, relational imbalance, and diminished self-integrity ([31]). Such students may therefore face an elevated risk of burnout, depressive symptoms, and difficulties in self-regulation due to their chronic overinvolvement in social relationships. Finally, the well-adjusted interpersonal group demonstrated low levels of interpersonal difficulties across dimensions, suggesting a balanced, flexible, and adaptive interpersonal style. In line with recent studies, these students were generally able to assert their needs, maintain appropriate boundaries, and cultivate mutually supportive relationships—hallmarks of healthy social functioning ([16]).

### 4.2. Network Structures of Adaptability and Mental Health

This study investigated the relationship between adaptability and mental health symptoms that vary across distinct interpersonal style groups among college students, leveraging a network analytic approach. The findings revealed a notable degree of structural invariance in the relationships between adaptability and mental health symptoms across the three interpersonal groups. Emotional adaptability and learning adaptability consistently showed strong negative associations with core indicators of mental health, including depression, anxiety, and stress. These results are consistent with previous studies highlighting adaptability as a protective psychological resource across different populations and contexts ([37]). However, the present findings extend this literature by demonstrating that the protective role of adaptability remains stable even when students are categorized into heterogeneous interpersonal style groups. This suggests that adaptability may operate as a transdiagnostic protective factor, exerting broad buffering effects that transcend variability in social functioning. In other words, although students differ in how they approach, maintain, or withdraw from interpersonal relationships, the foundational contribution of adaptability to mental health appears both robust and generalizable ([25]). This insight not only strengthens the empirical evidence base for adaptability as a core resilience factor but also underscores its practical relevance: adaptability-based interventions could serve as a universal strategy to enhance psychological well-being among college students, irrespective of their interpersonal style or disposition.

However, while the overall adaptability–mental health structure remained consistent, we observed meaningful group-specific differences within the internal organization of adaptability itself. Notably, in comparing the withdrawn and avoidant group to the well-adjusted interpersonal group, two edge-level disparities emerged: (1) the link between interpersonal adaptability and emotional adaptability, and (2) the connection between emotional adaptability and economic adaptability. In the well-adjusted interpersonal group, these adaptability components were tightly interconnected, suggesting that students in this group possess an integrated adaptability system that enables fluid transfer of self-regulatory skills across emotional, social, and practical domains. This finding is partially consistent with previous evidence showing that adaptive students are better able to generalize coping skills across contexts ([22]). In contrast, the withdrawn and avoidant group exhibited a noticeably fragmented adaptability network, indicating weaker integration among different adaptability components. Emotional adaptability appeared to be decoupled from other domains, suggesting that students in this group may struggle to apply emotional regulation skills beyond isolated situations. For example, they might manage their emotions in controlled or individual contexts but fail to transfer these strategies when facing interpersonal conflict or practical stressors such as financial difficulties.

### 4.3. Implications

The current findings yield several targeted implications for intervention and support strategies in higher education settings. First, while adaptability functions as a stable protective factor against psychological distress across interpersonal styles, students with a withdrawn and avoidant profile may remain at heightened risk due to the fragmented nature of their internal adaptability structure. For these students, interventions should move beyond general skills training and instead focus on fostering integration across emotional, interpersonal, and other contextual adaptabilities. Practical approaches may include cross-contextual role-playing exercises that help students transfer emotional regulation skills into interpersonal situations, or cognitive-behavioral techniques specifically designed to strengthen the connectivity between different adaptability domains. This recommendation is consistent with prior intervention studies emphasizing adaptability as a multi-domain construct ([28]; [25]), but our results further highlight the importance of targeting the linkages among adaptability dimensions, rather than treating them as isolated skills. Second, students in the overinvolved and compliant group may benefit from programs that focus on boundary-setting and the development of emotional autonomy. Such interventions could incorporate psychoeducational workshops on assertive communication or mindfulness-based strategies to reduce interpersonal overextension and prevent emotional exhaustion. These suggestions resonate with previous research that has linked interpersonal overinvolvement to burnout and depressive symptoms ([44]), and they reinforce the need for differentiated intervention strategies tailored to specific interpersonal vulnerabilities. Third, the results underscore the value of early screening during the transition to college, not only for interpersonal problems but also for adaptability coherence. By identifying students whose adaptability networks are fragmented or poorly integrated, institutions could design targeted, profile-specific support mechanisms before psychological distress escalates. Finally, network analysis simultaneously captures the relationship between adaptability and mental health dimensions and identifies both shared structures and group-specific differences through multi-group comparison. This approach uncovers underlying vulnerabilities, such as fragile connections between emotional and interpersonal adaptability, and guides interventions that enhance the total framework of adaptability instead of focusing on specific capabilities.

### 4.4. Limitation and Future Direction

This study has several limitations that warrant consideration. First, its cross-sectional design limits the ability to draw causal inferences about the dynamic relationship between adaptability and mental health symptoms across interpersonal profiles. Future research employing longitudinal designs or experience sampling methods is needed to capture the temporal dynamics and potential causal pathways within these psychological networks. Second, some subscales of the instruments, particularly the interpersonal problems measure, demonstrated relatively low internal consistency. This limitation was likely attributable to the small number of items within certain subscales, which constrained reliability and may have introduced measurement error. Our future studies should employ instruments with more appropriate item numbers or adopt alternative measures with better psychometric properties to improve the accuracy and validity of the assessment. Third, although the person-centered approach revealed theoretically meaningful subgroups, the generalizability of these interpersonal profiles may be constrained by the cultural and demographic homogeneity of the sample. In particular, the pronounced gender imbalance (with nearly 80% of participants being female) limits the extent to which these findings can be generalized, as gender roles and socialization processes may systematically shape interpersonal styles and adaptability. Moreover, the overall sample size was modest, which resulted in relatively small numbers of participants within each cluster. This limitation may have reduced the stability of network estimation and increased the likelihood of sampling variability in subgroup analyses. Replication in larger, more gender-balanced, and culturally diverse samples in the future is necessary to determine the robustness and applicability of these findings. Fourth, methodological constraints of the NCT should be acknowledged, as the method in this study only permits pairwise group comparisons and therefore increases the risk of inflated Type I error when multiple comparisons are conducted. Future methodological advances that enable simultaneous multi-group network comparisons would greatly improve the precision and efficiency of such analyses. Finally, the interpretation of betweenness centrality should be treated with caution. Prior research has shown that in psychological networks, where sample sizes are often modest and edge weights are estimated with uncertainty, betweenness tends to be unstable and highly sensitive to small data perturbations ([17]). Therefore, future research should employ larger sample sizes to further ensure the stability of centrality estimates in network analysis.

## 5. Conclusions

This study explored whether the associations between adaptability and mental health symptoms differ across students with distinct interpersonal styles. Using K-means clustering, three profiles were identified—withdrawn and avoidant, overinvolved and compliant, and well-adjusted—and their psychological networks were examined in depth. The results showed that group differences were subtle, largely limited to the interrelations among adaptability dimensions (e.g., emotional, interpersonal, and economic), rather than the direct links between adaptability and mental health symptoms. Across all profiles, emotional adaptability consistently emerged as a protective factor, negatively associated with depression, anxiety, and stress. These findings underscore the robust role of adaptability in promoting mental health. By integrating person-centered clustering with network analysis, this study offers a more nuanced understanding of how interpersonal functioning shapes students’ adjustment.

## Figures and Tables

**Figure 1 behavsci-15-01307-f001:**
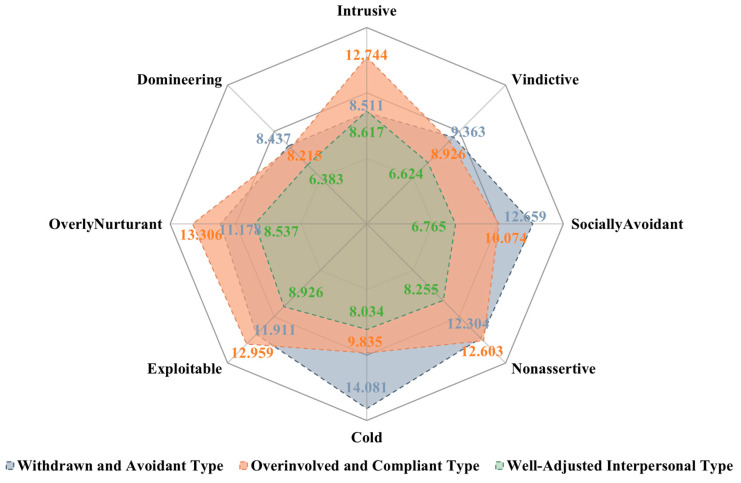
Mean scores on the dimensions of the inventory of interpersonal problems-circumplex scale across k-means clusters.

**Figure 2 behavsci-15-01307-f002:**
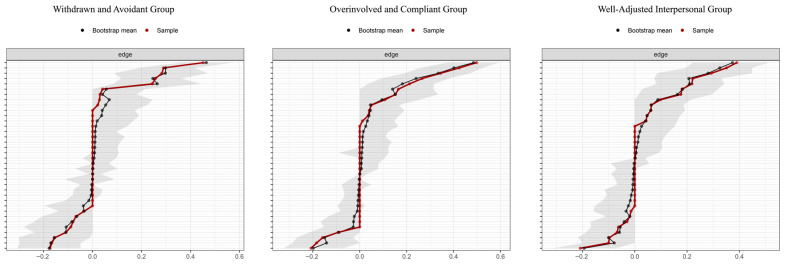
Network connection reliability between adaptability and mental health symptoms (Bootstrap n = 1000).

**Figure 3 behavsci-15-01307-f003:**
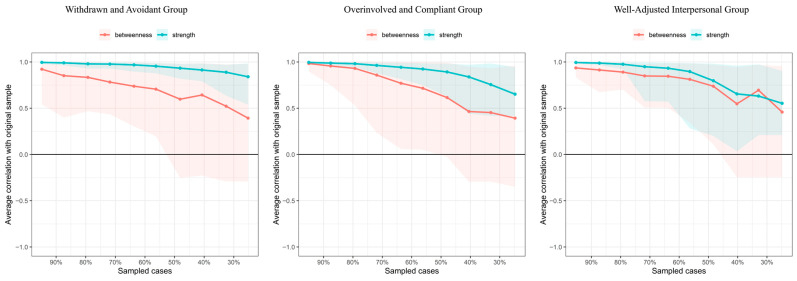
Stability of node centrality in the adaptability–mental health network (Bootstrap n = 1000).

**Figure 4 behavsci-15-01307-f004:**
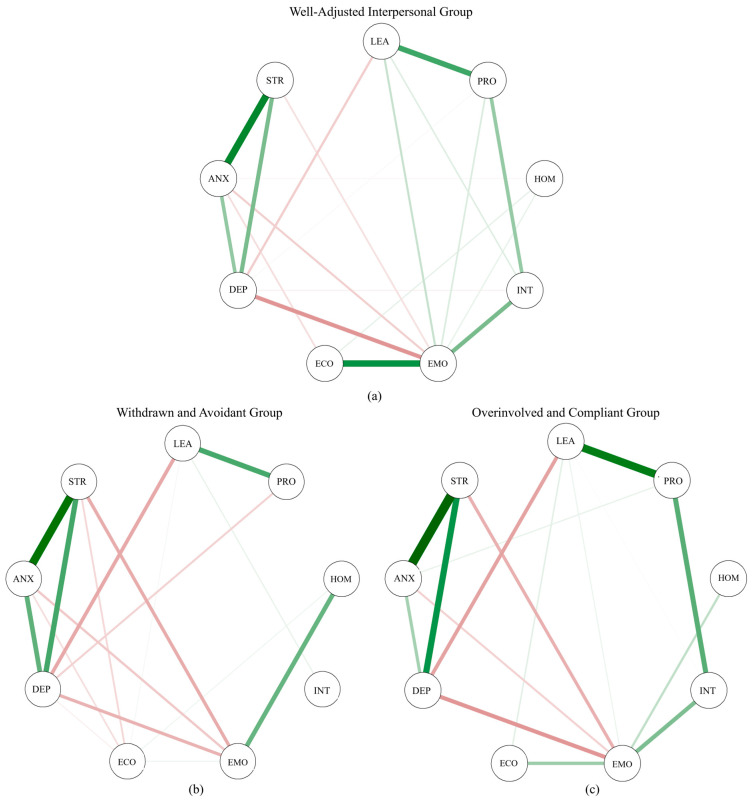
Network structure of mental health and adaptability for three interpersonal style groups: (**a**) Well-Adjusted interpersonal type; (**b**) Withdrawn and Avoidant type; (**c**) Overinvolved and Compliant type. Notes: (1) Positive connections between nodes are shown in green, and negative connections in red. Edge width reflects the strength of the association. (2) STR-stress; ANX-anxiety; DEP-depression; LEA-learning adaptability; PRO-professional adaptability; HOM-homesickness adaptability; INT-interpersonal adaptability; EMO-emotional adaptability; ECO-economic adaptability.

**Figure 5 behavsci-15-01307-f005:**
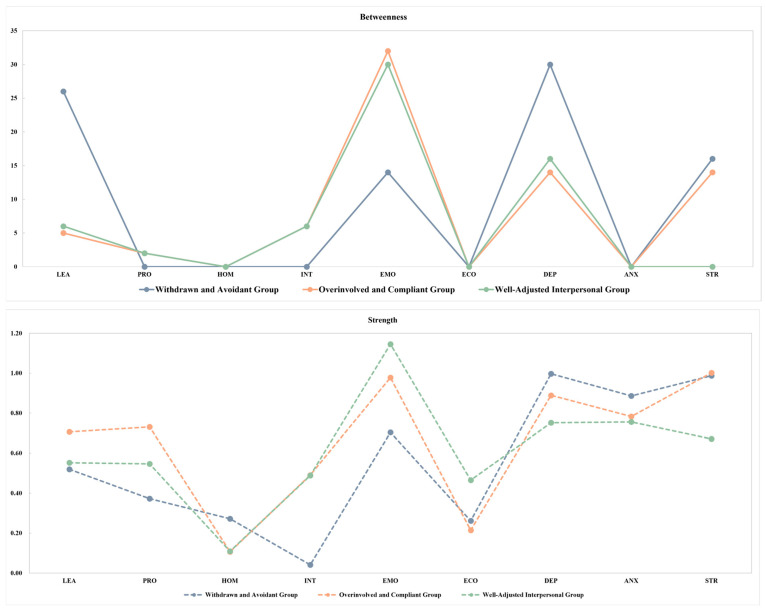
Node centrality in mental health and adaptability networks across interpersonal style groups. Notes: STR-stress; ANX-anxiety; DEP-depression; LEA-learning adaptability; PRO-professional adaptability; HOM-homesickness adaptability; INT-interpersonal adaptability; EMO-emotional adaptability; ECO-economic adaptability.

**Table 1 behavsci-15-01307-t001:** Indicators for different k-means clusters.

K	WCSS	CH	Dunn	BIC
2	2318.659	158.745	0.118	16,097.660
3	2052.483	115.510	0.118	16,030.080

## Data Availability

The raw data supporting the conclusions of this article will be made available by the authors on request.

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
