# Peer review of "Differential Associations Between Adaptability and Mental Health Symptoms Across Interpersonal Style Groups: A Network Comparison Study"

_behavsci, 2025, doi:10.3390/bs15101307_

Round 1

Reviewer 1 Report

Comments and Suggestions for Authors

This study addresses a timely and important topic with a sophisticated methodological approach. The integration of K-means clustering and psychological network analysis is commendable, and the manuscript is well-written. Below are suggested revisions to enhance the clarity, robustness, and interpretability of the manuscript:

  1. The decision to retain three interpersonal style clusters requires a stronger theoretical and empirical justification. This choice should be supported by statistical criteria, such as the Calinski–Harabasz index, as well as by reference to theory and validation of cluster stability.
  2. The conceptualization of interpersonal styles should be based on existing psychological models. The interpretive labels given to each cluster seem somewhat intuitive and post hoc. Relating these clusters to established frameworks, such as the interpersonal circumplex or attachment theory, would improve conceptual clarity.
  3. The statistical procedures used for network estimation and comparison must be described more explicitly. Specifically, clarify whether the data were standardized, if the regularization parameters were selected automatically, and how the choice of centrality indices was justified. Briefly acknowledge the limitations of betweenness centrality in psychological networks.
  4. Interpreting Network Comparison Test results requires more caution. While the manuscript emphasizes the structural differences between the two groups, it does not sufficiently discuss the limitations of statistical power or the practical significance of small edge differences. The finding of overall invariance in adaptability-mental health associations merits greater emphasis and theoretical consideration.
  5. The gender imbalance (almost 80% female) and the cultural specificity of the sample limit its generalizability. These limitations should be discussed more extensively, particularly with regard to how cultural norms and gender roles may influence interpersonal styles and adaptability processes.
  6. Integrating additional literature on psychological adjustment in university students would benefit the Discussion. Useful references to include are: 10.1080/00221325.2024.2413490; 10.1177/00332941221139713; 10.3389/fpsyt.2023.1329248.
Comments on the Quality of English Language

The manuscript would benefit from minor linguistic editing to enhance fluency and reduce repetitions, particularly in the Method section.

Author Response

# Response to Reviewer 1

This study addresses a timely and important topic with a sophisticated methodological approach. The integration of K-means clustering and psychological network analysis is commendable, and the manuscript is well-written. Below are suggested revisions to enhance the clarity, robustness, and interpretability of the manuscript:

1.The decision to retain three interpersonal style clusters requires a stronger theoretical and empirical justification. This choice should be supported by statistical criteria, such as the Calinski–Harabasz index, as well as by reference to theory and validation of cluster stability.

We sincerely appreciate you for this insightful suggestion. In the revised manuscript, we have provided a more detailed justification for retaining three interpersonal style clusters. Specifically, we re-described the clustering results and reported statistical criteria, including the within cluster sum of squares (WSS) and the Calinski-Harabasz index, to demonstrate that the three-cluster solution achieves a balance. In addition, following the reviewer’s advice, we have interpreted the three clusters from the perspective of attachment theory. This theoretical lens allowed us to clarify the psychological meaning of each cluster and to enhance the interpretability of the classification. It can be found in the “Results” section on Pages 9 and 10 of the revised manuscript. We have provided this information below as well, for your convenience:

“Table 1 presented the within cluster sum of squares (WSS) and Calinski–Harabasz (CH) indexes for different K-means clustering solutions, as determined using the Elbow method (Bholowalia & Kumar, 2018; Yock & Kim, 2017; Ashari et al., 2023). As expected, WSS decreased with increasing numbers of clusters. The most substantial improvement occurred when moving from two to three clusters (a reduction from 2318.659 to 2052.483), whereas the decrease from three to four clusters was marginal (to 1846.727). Consistently, the CH index indicated that both the two- and three-cluster solutions provided adequate separation (158.745 and 115.510, respectively), but dropped sharply at four clusters (100.267), suggesting over-partitioning. Taken together, these indices supported the three-cluster solution as providing the best balance between cohesion and separation while capturing meaningful heterogeneity in the data.

Figure 1 illustrated the mean scores across the dimensions of interpersonal problems for the three groups derived from the K-means clustering procedure. Cluster 1 (n = 135), labeled the “Withdrawn and Avoidant Type”, exhibited the highest scores on socially avoidant (M = 12.66), nonassertive (M = 12.30), and cold (M = 14.08), indicating a socially withdrawn and emotionally distant interpersonal style. Cluster 2 (n = 121), labeled the “Overinvolved and Compliant Type”, showed elevated scores on intrusive (M = 12.74), exploitable (M = 12.96), overly nurturant (M = 13.31), and nonassertive (M = 12.60), suggesting a pattern characterized by excessive self-disclosure, compliance, and a tendency to overextend in relationships. Cluster 3 (n = 149), labeled the “Well-Adjusted Interpersonal Type”, demonstrated comparatively lower scores across all dimensions, reflecting a more balanced and adaptive interpersonal style.

Beyond statistical fit, the three-cluster solution was also theoretically meaningful. The profiles closely resembled interpersonal style typologies frequently reported in previous research—namely, flexible–adaptive, exploitable–subservient, and hostile–avoidant (Wei et al., 2021). Moreover, they can be mapped onto established attachment theory (Berry et al., 2008): the well-adjusted interpersonal group corresponds to secure attachment, the overinvolved and compliant group reflects features of anxious–preoccupied attachment, and the withdrawn and avoidant group aligns with avoidant attachment (Bartholomew & Horowitz, 1991). Thus, the retention of three clusters was justified by both empirical robustness and theoretical grounding, enhancing the interpretability and validity of the findings.”

2.The conceptualization of interpersonal styles should be based on existing psychological models. The interpretive labels given to each cluster seem somewhat intuitive and post hoc. Relating these clusters to established frameworks, such as the interpersonal circumplex or attachment theory, would improve conceptual clarity.

We greatly appreciate the reviewer’s constructive feedback. In the revised manuscript, we have strengthened the conceptual grounding of the three interpersonal style clusters by explicitly linking them to attachment theory. Specifically, we interpreted the three clusters in relation to distinct attachment patterns, thereby enhancing the theoretical coherence of our classification. In addition, we updated the labels of the clusters to “Withdrawn and Avoidant Type,” “Overinvolved and Compliant Type,” and “Well-Adjusted Interpersonal Type” to improve clarity and align more closely with established psychological terminology. These revisions are presented in the “Results” section on Pages 9 and 10 of the revised manuscript.

3.The statistical procedures used for network estimation and comparison must be described more explicitly. Specifically, clarify whether the data were standardized, if the regularization parameters were selected automatically, and how the choice of centrality indices was justified. Briefly acknowledge the limitations of betweenness centrality in psychological networks.

We sincerely thank the reviewer for this helpful suggestion. In the revised manuscript, we have added a detailed description of the penalty parameter setting used in the EBICglasso network analysis. The corresponding explanation can be found in the “2.3 Software and Statistical Methods” section on Page 8 of the revised manuscript. We have provided this information below as well, for your convenience:

“…Subsequently, separate network models for mental health and adaptability were constructed for each subgroup via Gaussian Graphical Models (GGMs) using the R package qgraph (Epskamp et al., 2012). Network analysis, initially developed for sociological research, has recently gained prominence in psychological studies, where nodes represent psychological variables (e.g., adaptability, mental health symptoms) and edges represent their relationships (Di Blasi et al., 2021). To derive sparse partial correlation matrices emphasizing the most meaningful connections, the Extended Bayesian Information Criterion Graphical Least Absolute Shrinkage and Selection Operator (EBICGLASSO) algorithm was utilized (Epskamp et al., 2018). The current work established the tuning value λ, a hyperparameter that regulates the influence of the shrinkage penalty, at 0.5, in accordance with prior research (Zhang et al., 2023)…”

Furthermore, we have added the description about the limitations of betweenness centrality in psychological networks in this study. Details can be found on Pages 24 and 25. We have provided this information below as well, for your convenience:

“…Finally, the interpretation of betweenness centrality should be treated with caution. Prior research has shown that in psychological networks, where sample sizes are often modest and edge weights are estimated with uncertainty, betweenness tends to be unstable and highly sensitive to small data perturbations (Epskamp et al., 2018). Therefore, future research should employ larger sample sizes to further ensure the stability of centrality estimates in network analysis.”

4.Interpreting Network Comparison Test results requires more caution. While the manuscript emphasizes the structural differences between the two groups, it does not sufficiently discuss the limitations of statistical power or the practical significance of small edge differences. The finding of overall invariance in adaptability-mental health associations merits greater emphasis and theoretical consideration.

Thank you for this helpful suggestion. In the revised manuscript, we addressed this concern in two ways. First, for the three-group network comparison, we employed a more rigorous approach by increasing the number of permutations in the NCT from 1,000 to 2,000 when evaluating invariance in global strength, overall structure, and specific edges. While this reduces the likelihood of error, we acknowledge that the optimal solution would be the development of methods specifically designed for three-network comparisons. Accordingly, we have explicitly noted this limitation in the manuscript. These revisions are presented on Page 8 in the “2.3 Software and Statistical Methods” section and on Pages 24 and 25 in the “4.4 Limitations and Future Directions” section.

Furthermore, we have rewritten the Discussion section to place particular emphasis on the invariance of adaptability–mental health associations. The details can be found in the Discussion section on pages 21–22 of the revised manuscript.

5.The gender imbalance (almost 80% female) and the cultural specificity of the sample limit its generalizability. These limitations should be discussed more extensively, particularly with regard to how cultural norms and gender roles may influence interpersonal styles and adaptability processes.

Thank you for this important comment. We have now discussed the gender imbalance and cultural specificity of our sample in greater detail, emphasizing how cultural norms and gender roles may shape interpersonal styles and adaptability processes. This discussion has also been added to the limitations section of the revised manuscript (“4.4 Limitations and Future Directions”, pp. 24–25). We have provided this information below as well, for your convenience:

“…Third, although the person-centered approach revealed theoretically meaningful subgroups, the generalizability of these interpersonal profiles may be constrained by the cultural and demographic homogeneity of the sample. In particular, the pronounced gender imbalance (with nearly 80% of participants being female) limits the extent to which these findings can be generalized, as gender roles and socialization processes may systematically shape interpersonal styles and adaptability. Moreover, the overall sample size was modest, which resulted in relatively small numbers of participants within each cluster. This limitation may have reduced the stability of network estimation and increased the likelihood of sampling variability in subgroup analyses. Replication in larger, more gender-balanced, and culturally diverse samples in the future is necessary to determine the robustness and applicability of these findings. Fourth, methodological constraints of the NCT should be acknowledged that the method in this study only permits pairwise group comparisons and therefore increases the risk of inflated Type I error when multiple comparisons are conducted. Future methodological advances that enable simultaneous multi-group network comparisons would greatly improve the precision and efficiency of such analyses…”

6.Integrating additional literature on psychological adjustment in university students would benefit the Discussion. Useful references to include are: 10.1080/00221325.2024.2413490; 10.1177/00332941221139713; 10.3389/fpsyt.2023.1329248.

We appreciate this valuable suggestion. The recommended literature has been incorporated into the Discussion to better link our findings with previous research. The details can be found in the Discussion section on Pages 20–22.

7.The manuscript would benefit from minor linguistic editing to enhance fluency and reduce repetitions, particularly in the Method section.

Thank you for the suggestion. We have carefully revised the manuscript for grammar and style, with particular attention to the Method section, to improve fluency and reduce repetition.

Reviewer 2 Report

Comments and Suggestions for Authors

This study uses cluster analysis and network comparison methods to explore the differences in the associations between adaptability and mental health symptoms among college students with different interpersonal style groups. The researchers identified three interpersonal style groups through K-means clustering: withdrawn and passive, overinvolved and self-sacrificing, and well-adjusted. Based on this, network models were constructed to analyze the relationships between six dimensions of adaptability  and three core mental health symptoms.

Overall, I believe the paper is unlikely to meet the standards for publication. Below are suggestions for revision.

First, there are many unscientific aspects in the use of scales and methods, which are very unrigorous.

Second, the paper uses the k=3 solution in K-means clustering, but the Calinski-Harabasz index is highest at k=2 (158.745), while k=3 (115.510) is relatively high but not optimal. Moreover, the decision is based on the reduction of within-cluster sum of squares, which is a subjective criterion, introducing the risk of clustering instability.

Third, the total sample size of 405 is divided into three groups (n=135, 121, 149), with each group having a relatively small sample size, especially the overinvolved and self-sacrificing group (n=121), where the network stability coefficient (node strength centrality stability coefficient is only 0.438) is close to the critical value of 0.25, leading to estimation errors in edge weights.

Fourth, some dimensions of the scales have low reliability (e.g., some dimensions of the interpersonal problems scale have Cronbach's α = 0.554).

Fifth, the network comparison method has limitations. NCT only supports pairwise group comparisons and cannot analyze global differences among three groups simultaneously; moreover, 1,000 permutation tests may be insufficient to detect subtle structural changes.

Comments on the Quality of English Language

The English is poor; it is recommended to have it polished by a professional. 

Author Response

# Response to Reviewer 2

Comment 1: This study uses cluster analysis and network comparison methods to explore the differences in the associations between adaptability and mental health symptoms among college students with different interpersonal style groups. The researchers identified three interpersonal style groups through K-means clustering: withdrawn and passive, overinvolved and self-sacrificing, and well-adjusted. Based on this, network models were constructed to analyze the relationships between six dimensions of adaptability and three core mental health symptoms. Overall, I believe the paper is unlikely to meet the standards for publication. Below are suggestions for revision.

First, there are many unscientific aspects in the use of scales and methods, which are very unrigorous.

Response 1: We sincerely thank the reviewer for this critical observation. In conducting this study, we made every effort to select instruments with established reliability and validity to ensure methodological rigor. Nevertheless, we acknowledge that one dimension of the interpersonal problem scale exhibited relatively low reliability, which is likely attributable to the limited number of items in that subscale. We fully recognize this as a limitation of the present study and have explicitly added a discussion of this issue in the “4.4 Limitations and Future Directions” section (pp. 24–25). We hope this clarification demonstrates our careful consideration of measurement issues and our commitment to transparency in reporting.

Comment 2: Second, the paper uses the k=3 solution in K-means clustering, but the Calinski-Harabasz index is highest at k=2 (158.745), while k=3 (115.510) is relatively high but not optimal. Moreover, the decision is based on the reduction of within-cluster sum of squares, which is a subjective criterion, introducing the risk of clustering instability.

Response 2: We sincerely appreciate you for this insightful suggestion. In the revised manuscript, we have provided a more detailed justification for retaining three interpersonal style clusters. Specifically, we re-described the clustering results and reported statistical criteria, including the within cluster sum of squares (WSS) and the Calinski-Harabasz index, to demonstrate that the three-cluster solution achieves a balance. In addition, following the reviewer’s advice, we have interpreted the three clusters from the perspective of attachment theory. This theoretical lens allowed us to clarify the psychological meaning of each cluster and to enhance the interpretability of the classification. It can be found in the “Results” section on Pages 9 and 10 of the revised manuscript. We have provided this information below as well, for your convenience:

“Table 1 presented the within cluster sum of squares (WSS) and Calinski–Harabasz (CH) indexes for different K-means clustering solutions, as determined using the Elbow method (Bholowalia & Kumar, 2018; Yock & Kim, 2017; Ashari et al., 2023). As expected, WSS decreased with increasing numbers of clusters. The most substantial improvement occurred when moving from two to three clusters (a reduction from 2318.659 to 2052.483), whereas the decrease from three to four clusters was marginal (to 1846.727). Consistently, the CH index indicated that both the two- and three-cluster solutions provided adequate separation (158.745 and 115.510, respectively), but dropped sharply at four clusters (100.267), suggesting over-partitioning. Taken together, these indices supported the three-cluster solution as providing the best balance between cohesion and separation while capturing meaningful heterogeneity in the data.

Figure 1 illustrated the mean scores across the dimensions of interpersonal problems for the three groups derived from the K-means clustering procedure. Cluster 1 (n = 135), labeled the “Withdrawn and Avoidant Type”, exhibited the highest scores on socially avoidant (M = 12.66), nonassertive (M = 12.30), and cold (M = 14.08), indicating a socially withdrawn and emotionally distant interpersonal style. Cluster 2 (n = 121), labeled the “Overinvolved and Compliant Type”, showed elevated scores on intrusive (M = 12.74), exploitable (M = 12.96), overly nurturant (M = 13.31), and nonassertive (M = 12.60), suggesting a pattern characterized by excessive self-disclosure, compliance, and a tendency to overextend in relationships. Cluster 3 (n = 149), labeled the “Well-Adjusted Interpersonal Type”, demonstrated comparatively lower scores across all dimensions, reflecting a more balanced and adaptive interpersonal style.

Beyond statistical fit, the three-cluster solution was also theoretically meaningful. The profiles closely resembled interpersonal style typologies frequently reported in previous research—namely, flexible–adaptive, exploitable–subservient, and hostile–avoidant (Wei et al., 2021). Moreover, they can be mapped onto established attachment theory (Berry et al., 2008): the well-adjusted interpersonal group corresponds to secure attachment, the overinvolved and compliant group reflects features of anxious–preoccupied attachment, and the withdrawn and avoidant group aligns with avoidant attachment (Bartholomew & Horowitz, 1991). Thus, the retention of three clusters was justified by both empirical robustness and theoretical grounding, enhancing the interpretability and validity of the findings.”

Comment 3: Third, the total sample size of 405 is divided into three groups (n=135, 121, 149), with each group having a relatively small sample size, especially the overinvolved and self-sacrificing group (n=121), where the network stability coefficient (node strength centrality stability coefficient is only 0.438) is close to the critical value of 0.25, leading to estimation errors in edge weights.

Response 3: We thank the reviewer for highlighting this important concern. We fully acknowledge that the relatively small sample size in each subgroup may have affected the stability of the network estimation. We have now explicitly discussed this issue in the “4.4 Limitations and Future Directions” section (pp. 24–25) of the revised manuscript. We have provided this information below as well, for your convenience:

“This study has several limitations that warrant consideration. First, its cross-sectional design limits the ability to draw causal inferences about the dynamic relationship between adaptability and mental health symptoms across interpersonal profiles. Future research employing longitudinal designs or experience sampling methods is needed to capture the temporal dynamics and potential causal pathways within these psychological networks. Second, some subscales of the instruments, particularly the interpersonal problems measure, demonstrated relatively low internal consistency. This limitation was likely attributable to the small number of items within certain subscales, which constrained reliability and may have introduced measurement error. Our future studies should employ instruments with more appropriate item numbers or adopt alternative measures with better psychometric properties to improve the accuracy and validity of assessment. Third, although the person-centered approach revealed theoretically meaningful subgroups, the generalizability of these interpersonal profiles may be constrained by the cultural and demographic homogeneity of the sample. In particular, the pronounced gender imbalance (with nearly 80% of participants being female) limits the extent to which these findings can be generalized, as gender roles and socialization processes may systematically shape interpersonal styles and adaptability. Moreover, the overall sample size was modest, which resulted in relatively small numbers of participants within each cluster. This limitation may have reduced the stability of network estimation and increased the likelihood of sampling variability in subgroup analyses. Replication in larger, more gender-balanced, and culturally diverse samples in the future is necessary to determine the robustness and applicability of these findings. Fourth, methodological constraints of the NCT should be acknowledged that the method in this study only permits pairwise group comparisons and therefore increases the risk of inflated Type I error when multiple comparisons are conducted. Future methodological advances that enable simultaneous multi-group network comparisons would greatly improve the precision and efficiency of such analyses. Finally, the interpretation of betweenness centrality should be treated with caution. Prior research has shown that in psychological networks, where sample sizes are often modest and edge weights are estimated with uncertainty, betweenness tends to be unstable and highly sensitive to small data perturbations (Epskamp et al., 2018). Therefore, future research should employ larger sample sizes to further ensure the stability of centrality estimates in network analysis.”

Comment 4: Fourth, some dimensions of the scales have low reliability (e.g., some dimensions of the interpersonal problems scale have Cronbach's α = 0.554).

Response 4: We sincerely thank the reviewer for pointing out this important issue. Indeed, we acknowledge that one dimension of the interpersonal problems scale demonstrated relatively low internal consistency. We have now explicitly addressed this limitation in the “4.4 Limitations and Future Directions” section (pp. 24–25). While this scale has been widely used in previous research, the relatively low reliability of this dimension suggests that its psychometric properties warrant further refinement. Future studies should therefore consider re-examining this dimension, potentially expanding the number of items or revising their content to improve measurement reliability.

Comment 5: Fifth, the network comparison method has limitations. NCT only supports pairwise group comparisons and cannot analyze global differences among three groups simultaneously; moreover, 1,000 permutation tests may be insufficient to detect subtle structural changes.

Response 5: Thank you for this helpful suggestion. In the revised manuscript, we adopted a more rigorous approach by increasing the number of permutations in the NCT from 1,000 to 2,000 when evaluating invariance in global strength, overall structure, and specific edges. This adjustment resulted in only minor numerical fluctuations, with no substantive change in the conclusions. The detailed revisions are presented in the “2.3 Software and Statistical Methods” section (p. 8) and the “Results” section (pp. 19).

Comment 6: Sixth, the English is poor; it is recommended to have it polished by a professional.

Response 6: Thank you for the suggestion. We have carefully revised the manuscript for grammar and style to improve fluency and reduce repetition.

Reviewer 3 Report

Comments and Suggestions for Authors

Review Comments:

Thank you for the opportunity to review this manuscript. This study examines how the associations between adaptability and mental health symptoms vary across distinct interpersonal style profiles. The manuscript is generally well written, and the topic is both highly relevant and interesting. Moreover, the use of K-means clustering and network comparison tests represents a methodological strength, yielding novel and insightful findings. I recommend that the manuscript be accepted for publication pending minor revisions, as outlined below.

1. The text in Figures 2 and 3 is too small to be read comfortably. Please adjust the font size to improve legibility.

2. In the discussion and implication sections, there is no comparison of the present results with previous findings. The authors should strengthen the validation of their results by referring to the relevant literature.

I wish the authors continued success in their research.

Author Response

# Response to Reviewer 3

Comments and Suggestions for Authors

Review Comments:

Thank you for the opportunity to review this manuscript. This study examines how the associations between adaptability and mental health symptoms vary across distinct interpersonal style profiles. The manuscript is generally well written, and the topic is both highly relevant and interesting. Moreover, the use of K-means clustering and network comparison tests represents a methodological strength, yielding novel and insightful findings. I recommend that the manuscript be accepted for publication pending minor revisions, as outlined below.

Comment 1: The text in Figures 2 and 3 is too small to be read comfortably. Please adjust the font size to improve legibility.

Thank you for this helpful comment. In the revised manuscript, the font size in Figures 2 and 3 has been increased to enhance clarity and readability. These revisions can be found on Pages 14 and 15 of the revised manuscript.

Comment 2: In the discussion and implication sections, there is no comparison of the present results with previous findings. The authors should strengthen the validation of their results by referring to the relevant literature.

We sincerely thank you for this valuable suggestion. In the revised manuscript, we have revised the Discussion section to provide a clearer comparison between our findings and previous research. This suggestion has been particularly helpful in enhancing the depth of our discussion, and we are grateful once again for the reviewer’s insightful comment. Details can be seen on Pages 20, 21, 22, and 23.

3. I wish the authors continued success in their research.

We sincerely thank the reviewer for the encouraging comment and kind wishes. We truly appreciate your support and will continue to advance our research in this area.

Reviewer 4 Report

Comments and Suggestions for Authors

Overall comments:

The paper titled Differential Associations Between Adaptability and Mental Health Symptoms Across Interpersonal Style Groups: A Network Comparison Study” presents a study that aims to explore how the relations between adaptability and mental health symptoms vary across distinct interpersonal style profiles. This research paper makes very very significant contribution to the field of understanding the interplay between interpersonalč styles, mental health, and adaptability in students with so many great practice implications. In addition, it applied very contemporary methodology and statistics which can serve fine for future similar studies.  Therefore, it has great potential for publishing. However, the paper has some parts that require revision, so after the necessary revisions are made, the paper can be considered for publication.

Abstract: The beginning of the abstract is not clear, and this may be some kind of typing error – when you wrote university is a transitional period, you probably meant university period or studying period, so please rewrite this.

Introduction: This part of the paper is written very well. However, two parts are missing – the theoretical background on mental health, i.e., aspects you measured such as depression, anxiety and stress, so please describe the basic theoretical background on mental health in this paper. In addition, the theoretical background on the concept of adaptability is missing, too, so please describe it in detail. Also, both theoretical models should be accompanied by contemporary and relevant references.

Methods: Participants: in this part of the paper, an explanation of the type of sample is missing, so please describe and name it. Furthermore, the subdivision of Procedure is missing (some parts are written within subdivision Participants, but you should place it within Procedure with the addition of a detailed research procedure from recruiting participants, over the duration of their time for filling the scales. Very important, how you managed ethical challenges regarding the application of DASS, since it has some potentially disturbing questions for those participants who are in the group of experiencing anxiety and depression, and stress more frequently than others. How do you ensure for them to feel ok after filling the scale with potentially disturbing items/questions? Measurements: The inventory of interpersonal problems circumplex scale: how you explain determined range from .554 to .787 of Cronbach's alphas for this scale’s subscales. The Cronbach’s α of .554 is very low – please explain, and if necessary, include this in your data interpretation along with the study limitations.

Discussion: This part should be more critical of the determined results regarding the previous results from other studies – there is no comparison with other research findings here, so please revise this accordingly.

Literature: The literature is contemporary enough.

Conclusion: The paper should be revised according to the suggestions.

Author Response

# Response to Reviewer 4

Overall comments:

Comment 1: The paper titled Differential Associations Between Adaptability and Mental Health Symptoms Across Interpersonal Style Groups: A Network Comparison Study” presents a study that aims to explore how the relations between adaptability and mental health symptoms vary across distinct interpersonal style profiles. This research paper makes very significant contribution to the field of understanding the interplay between interpersonal styles, mental health, and adaptability in students with so many great practice implications. In addition, it applied very contemporary methodology and statistics which can serve fine for future similar studies.  Therefore, it has great potential for publishing. However, the paper has some parts that require revision, so after the necessary revisions are made, the paper can be considered for publication.

Abstract: The beginning of the abstract is not clear, and this may be some kind of typing error – when you wrote university is a transitional period, you probably meant university period or studying period, so please rewrite this.

Response 1: Thanks for your carefully review. We have revised the beginning of the abstract to improve clarity and correct the wording.

Comment 2:

Introduction: This part of the paper is written very well. However, two parts are missing – the theoretical background on mental health, i.e., aspects you measured such as depression, anxiety and stress, so please describe the basic theoretical background on mental health in this paper. In addition, the theoretical background on the concept of adaptability is missing, too, so please describe it in detail. Also, both theoretical models should be accompanied by contemporary and relevant references.

Response 2: We sincerely thank the reviewer for this constructive suggestion. In the revised manuscript, we have enriched the Introduction by adding detailed explanations and definitions of both adaptability and mental health (including depression, anxiety, and stress). These additions can be found in the “Introduction” section on Pages 2 and 3 of the revised manuscript. We have provided this information below as well, for your convenience:

“…A growing body of research has underscored the strong association between college students’ interpersonal functioning and mental health (Ryan et al., 2023; Lee et al., 2021). Mental health is commonly conceptualized as a multidimensional construct and a dynamic state of internal equilibrium that enables individuals to realize their abilities in harmony with the universal values of society (Galderisi et al., 2017). Within the university context, depression, anxiety, and stress are among the most prevalent and impairing forms of mental health, as they are closely linked to academic demands, social transitions, and identity development (Ali et al., 2021; Moya et al., 2022). Importantly, prior studies have shown that interpersonal functioning plays an important role in shaping these outcomes. For instance, Yuan et al. (2022) found that students exhibiting high levels of social avoidance tend to report lower self-esteem and reduced interpersonal trust, which in turn contribute to elevated symptoms of depression and anxiety. In contrast, students who demonstrate adaptive interpersonal behaviors are more likely to experience fewer mental health problems (Black et al., 2019; Shin & Newman, 2019).

Beyond its impact on mental health, interpersonal functioning also plays a vital role in shaping students’ adaptability to both college life and the broader social environment (Leary & DeRosier, 2012; Kim, 2022; Zhang et al., 2021). Adaptability, in turn, is recognized as a crucial psychological capacity that enables students to adjust their thoughts, emotions, and behaviors in response to dynamic and uncertain circumstances (Cao & Mao, 2008; Luo, 2020). It encompasses multiple domains, including learning adaptability, professional adaptability, homesickness adaptability, interpersonal adaptability, emotional adaptability, and economic adaptability. When adaptability is low, maladaptive interpersonal patterns may further exacerbate adjustment difficulties. For example, Gomez Penedo et al. (2019) reported that individuals characterized by excessive hostility or dependency often struggle with emotional maladjustment. Similarly, those with social avoidance tendencies are more likely to experience challenges such as homesickness and difficulties in interpersonal adjustment following college enrollment (Ren et al., 2024; Rathakrishnan et al., 2021) …”

Comment 3:

Methods: Participants: in this part of the paper, an explanation of the type of sample is missing, so please describe and name it. Furthermore, the subdivision of Procedure is missing (some parts are written within subdivision Participants, but you should place it within Procedure with the addition of a detailed research procedure from recruiting participants, over the duration of their time for filling the scales. Very important, how you managed ethical challenges regarding the application of DASS, since it has some potentially disturbing questions for those participants who are in the group of experiencing anxiety and depression, and stress more frequently than others. How do you ensure for them to feel ok after filling the scale with potentially disturbing items/questions?

Response 3: We sincerely thank you for this valuable suggestion. In the revised manuscript, we have specified the type of sample in the Participants section and added a separate Procedure section. Importantly, we have also added an explanation of how we managed the ethical considerations related to the DASS in this section, including measures taken to ensure that participants felt supported and comfortable after responding to potentially sensitive items. Details can be found on Pages 5 and 6. We have provided this information below as well, for your convenience:

“We recruited a convenience sample of 405 undergraduate and graduate students from Tianjin, a large city in China. Participants were invited through classroom announcements and voluntarily took part in the study. The survey was administered in paper–pencil format and took approximately 5–8 minutes to complete. Data were collected in groups of about 90 individuals. To examine the extent of missing data, univariate statistical analyses were performed on each item. Results indicated that missing data rates ranged from 0.0% to 1.2%. As missingness was below the 10.0% threshold, and different handling methods did not yield significant differences (Newman, 2014), the expectation–maximization (EM) imputation method was applied. The final sample consisted of 405 participants, with 70 males (20.8%) and 335 females (79.2%), with an average age of 19.520 years (SD = 1.932).”

“Participants were informed of the study purpose, confidentiality, and their right to withdraw at any point prior to data collection. Written informed consent was obtained from all participants before completing the survey. The study was reviewed and approved by the Human Research Protection Committee of Tianjin Normal University (ethical approval number: XL2020-08). Special attention was given to the administration of the Depression Anxiety Stress Scales (DASS-21), as it contains potentially distressing items. To minimize risks, participants were informed in advance that some questions might involve sensitive emotional content, but were reassured that there were no right or wrong answers. After completing the survey, all participants were debriefed to ensure they felt comfortable and safe.”

Comment 4:

Measurements: The inventory of interpersonal problems circumplex scale: how you explain determined range from .554 to .787 of Cronbach's alphas for this scale’s subscales. The Cronbach’s α of .554 is very low – please explain, and if necessary, include this in your data interpretation along with the study limitations.

Response 4: We sincerely thank the reviewer for this critical observation. In conducting this study, we made every effort to select instruments with established reliability and validity to ensure methodological rigor. Nevertheless, we acknowledge that one dimension of the interpersonal problem scale exhibited relatively low reliability, which is likely attributable to the limited number of items in that subscale. We fully recognize this as a limitation of the present study and have explicitly added a discussion of this issue in the “4.4 Limitations and Future Directions” section (pp. 24–25). We hope this clarification demonstrates our careful consideration of measurement issues and our commitment to transparency in reporting.

Comment 5:

Discussion: This part should be more critical of the determined results regarding the previous results from other studies – there is no comparison with other research findings here, so please revise this accordingly.

Response 5: We sincerely thank you for this valuable suggestion. In the revised manuscript, we have revised the Discussion section to provide a clearer comparison between our findings and previous research. This suggestion has been particularly helpful in enhancing the depth of our discussion, and we are grateful once again for the reviewer’s insightful comment. Details can be seen on Pages 20, 21, 22, and 23.

Comment 5:

Literature: The literature is contemporary enough.

Response 5: We thank the reviewer for this positive comment and appreciate the recognition of the relevance and currency of the literature used in our manuscript.

Comment 6:

Conclusion: The paper should be revised according to the suggestions.

Response 6: Thanks for your helpful suggestion. In the revised manuscript, we have added a separate Conclusion section, which can be found on page 25. We have provided this information below as well, for your convenience:

“This study explored whether the associations between adaptability and mental health symptoms differ across students with distinct interpersonal styles. Using K-means clustering, three profiles were identified—withdrawn and avoidant, overinvolved and compliant, and well-adjusted—and their psychological networks were examined in depth. The results showed that group differences were subtle, largely limited to the interrelations among adaptability dimensions (e.g., emotional, interpersonal, economic), rather than the direct links between adaptability and mental health symptoms. Across all profiles, emotional adaptability consistently emerged as a protective factor, negatively associated with depression, anxiety, and stress. These findings underscore the robust role of adaptability in promoting mental health. By integrating person-centered clustering with network analysis, this study offers a more nuanced understanding of how interpersonal functioning shapes students’ adjustment.”

Round 2

Reviewer 1 Report

Comments and Suggestions for Authors

Thank you for the thorough revisions. The manuscript is now clearer, methodologically solid, and better integrated with the current literature. 

Overall, the paper is much improved and nearly ready for publication. Congratulations!

Author Response

Thank you for the thorough revisions. The manuscript is now clearer, methodologically solid, and better integrated with the current literature. 

Overall, the paper is much improved and nearly ready for publication. Congratulations!

Response: Thank you for your time and consideration of our manuscript. We appreciate your feedback and are glad that no further revisions were required based on your review.

Reviewer 2 Report

Comments and Suggestions for Authors

I have carefully read the revised draft and your responses to various issues, but unfortunately, I still believe that there are fundamental flaws in the methodology of the paper.

The author acknowledged that the reliability of some key scales was too low (Cronbach's α = 0.554) and listed it as a research limitation. A scale with such a low reliability indicates a significant measurement error, and all subsequent analyses based on this data are unreliable. Merely mentioning it in the "Limitations" section cannot make up for the fundamental defect in data quality.

Although the author added a discussion on clustering indicators, the statistical indicators showed that the optimal clustering scheme was k=2. However, the author still insisted on using the k=3 scheme and mainly based it on theoretical interpretability. This indicates that the researcher abandoned the objective statistical standards in order to verify the preset theory. Such a practice violates the research rigor and makes the clustering results themselves untrustworthy.

The author also acknowledges that due to the relatively small subsample size, the network stability coefficient is close to the critical value, which makes the estimation of the network model unstable. Under such circumstances, the results of network comparisons based on these unstable models are unreliable, and thus no robust conclusions can be drawn.

Author Response

# Response to Reviewer 2

I have carefully read the revised draft and your responses to various issues, but unfortunately, I still believe that there are fundamental flaws in the methodology of the paper.

  1. The author acknowledged that the reliability of some key scales was too low (Cronbach's α = 0.554) and listed it as a research limitation. A scale with such a low reliability indicates a significant measurement error, and all subsequent analyses based on this data are unreliable. Merely mentioning it in the "Limitations" section cannot make up for the fundamental defect in data quality.

We sincerely appreciate your valuable suggestion. For the Cronbach’s α = 0.554 of exploitable dimension, we acknowledge that is not particularly high; however, this lower value is largely attributable to the small number of items in this subscale, a factor that has been widely recognized in both methodological and empirical research as acceptable under such circumstances (Komorita & Graham, 1965; Abdelmoula et al., 2015; Luh, 2024). For example, Komorita and Graham (1965) emphasized that “the magnitude of the coefficient alpha is contingent upon the number of items, with a curvilinear relationship”, while Abdelmoula (2015) reported that “there is a consensus in the literature that the value of alpha coefficient depends on the number of items.” Moreover, previous studies have demonstrated that subscales with lower reliability due to limited items can still be scientifically useful. For instance, Bernal et al. (2003) reported a subscale with α = 0.59 that was nonetheless widely applied; Daemen et al. (2022) found reliabilities as low as 0.2–0.7 for momentary self-esteem scale; and Pavlickova (2015) reported an α of 0.5 for a self-esteem scale. These examples collectively demonstrate that reduced reliability resulting from fewer items does not necessarily invalidate the utility of a measure.

Considering this evidence, while we acknowledge the limitation of the exploitable dimension, we believe that—supported by theoretical justification, acceptable reliabilities of the other dimensions, and precedents in the literature—the instrument remains appropriate for research use. In the revision, we have added more supporting references and will emphasize this limitation with greater caution in the discussion. Detailed revisions can be found on Page 6 and 23 of the revised manuscript.

Abdelmoula, M., Chakroun, W., & Akrout, F. (2015). The effect of sample size and the number of items on reliability coefficients: Alpha and rho: A meta-analysis. International Journal of Numerical Methods and Applications, 13(1), 1-20. http://dx.doi.org/10.17654/IJNMAMar2015_001_020

Bernal, G., Molina, M. M. M., & del Río, M. R. S. (2003). Development of a brief scale for social support: Reliability and validity in Puerto Rico. International Journal of Clinical and Health Psychology, 3(2), 251-264. https://www.redalyc.org/pdf/337/33730203.pdf

Daemen, M., van Amelsvoort, T., GROUP Investigators, & Reininghaus, U. (2022). Self-esteem and psychosis in daily life: An experience sampling study. Journal of Psychopathology and Clinical Science, 131(2), 182–197. https://doi.org/10.1037/abn0000722

Komorita, S. S., & Graham, W. K. (1965). Number of scale points and the reliability of scales. Educational and Psychological Measurement, 25(4), 987–995. https://doi.org/10.1177/001316446502500404

Luh, W. M. (2024). A general framework for planning the number of items/subjects for evaluating Cronbach’s alpha: Integration of hypothesis testing and confidence intervals. Methodology, 20(1), 1-21. https://doi.org/10.5964/meth.10449

Pavlickova, H., Turnbull, O. H., Myin-Germeys, I., & Bentall, R. P. (2015). The inter-relationship between mood, self-esteem and response styles in adolescent offspring of bipolar parents: an experience sampling study. Psychiatry research, 225(3), 563-570. https://doi.org/10.1016/j.psychres.2014.11.046

  1. Although the author added a discussion on clustering indicators, the statistical indicators showed that the optimal clustering scheme was k=2. However, the author still insisted on using the k=3 scheme and mainly based it on theoretical interpretability. This indicates that the researcher abandoned the objective statistical standards in order to verify the preset theory. Such a practice violates the research rigor and makes the clustering results themselves untrustworthy.

Thank you for your valuable comment. It should be clarified that in K-means clustering, no single index can be regarded as a definitive “gold standard” for determining the number of clusters. Instead, clustering solutions should be evaluated in conjunction with relevant psychological theory. This view has been consistently highlighted in the literature. For instance, Schubert (2023) argued that “there is no ‘optimal’ solution in cluster analysis, but it is an explorative approach that may yield multiple interesting solutions, and interestingness necessarily is a subjective decision of the user.” Consistent with this perspective, we extended our evaluation beyond the indices previously reported by incorporating the within-cluster sum of squares (WCSS), the Calinski–Harabasz (CH) index, the Dunn index, and the Bayesian Information Criterion (BIC) (Kodinariya & Makwana, 2013). The CH index favored a two-cluster solution, the Dunn index indicated that both two and three clusters were plausible, while the BIC supported three clusters. Details can be seen in Table 1. We also added this information into the revised version of the manuscript. It can be located on Pages 9 and 10 in the revised manuscript.

Table 1

Indies for different k-means clusters

K

WCSS

CH

Dunn

BIC

2

2318.659

158.745

0.118

16097.660

3

2052.483

115.510

0.118

16030.080

We also conducted Latent Profile Analysis (LPA) as an alternative classification method, which likewise favored a three-class model, as shown below in Table 2:

Table 2

The indexes of model fit for LPA

AIC

BIC

aBIC

2 classes

15923.59

16023.69

15944.36

3 classes

15776.46

15912.6

15804.71

Finally, we considered attachment theory, which identifies three primary attachment styles in close relationships—secure, anxious–preoccupied, and avoidant. Taken together, the theoretical rationale, multiple clustering indices, and the LPA validation provide converging evidence for selecting the three-cluster solution in this study.

Schubert, E. (2023). Stop using the elbow criterion for k-means and how to choose the number of clusters instead. ACM SIGKDD Explorations Newsletter, 25(1), 36-42. https://doi.org/10.1145/3606274.3606278

Kodinariya, T. M., & Makwana, P. R. (2013). Review on determining number of Cluster in K-Means Clustering. International Journal, 1(6), 90-95.

  1. The author also acknowledges that due to the relatively small subsample size, the network stability coefficient is close to the critical value, which makes the estimation of the network model unstable. Under such circumstances, the results of network comparisons based on these unstable models are unreliable, and thus no robust conclusions can be drawn.

We thank for you highlighting this important concern. We acknowledge that there is no universally agreed-upon standard for the minimum sample size required in psychological network analysis. Nevertheless, prior work provides useful guidance. Epskamp and Fried (2018) emphasized that for EBICglasso to produce stable estimates, the sample size (?) should be substantially larger than the number of variables (?). In our study, ? was nine and ? exceeded 100 in each subgroup, which meets this recommended criterion.

Moreover, following the recommendations of Epskamp, Rhemtulla, and Borsboom (2017), who advised that bootstrap procedures can be employed to evaluate stability in smaller samples, we conducted 1,000 bootstrap iterations to obtain confidence intervals for edge weights. This allowed us to directly assess the reliability of the estimated connections. We also applied case-dropping bootstraps using the bootnet package to examine the stability of network structures across subgroups. These analyses consistently indicated that the estimated networks were stable.

Finally, for the three-group network comparison, we adopted a more stringent procedure by increasing the number of permutations in the Network Comparison Test (NCT) from 1,000 to 2,000 when evaluating invariance in global strength, overall structure, and specific edges. This adjustment resulted in only minor numerical fluctuations and did not alter the substantive conclusions. The detailed procedures and results are reported in the Methods and Results sections of the revised manuscript.

Epskamp, S., & Fried, E. I. (2018). A tutorial on regularized partial correlation networks. Psychological Methods, 23(4), 617–634. https://doi.org/10.1037/met0000167

Epskamp, S., Rhemtulla, M., & Borsboom, D. (2017). Estimating psychological networks and their accuracy: A tutorial paper. Behavior Research Methods, 50, 195–212. https://doi.org/10.3758/s13428-017-0862-1

Reviewer 4 Report

Comments and Suggestions for Authors

Thank you for revising your manuscript according to suggestions - now it is ready for publication

Author Response

Thank you for revising your manuscript according to suggestions - now it is ready for publication.

Response: Thank you for your time and consideration of our manuscript. We appreciate your feedback and are glad that no further revisions were required based on your review.